# Recovery of europium from E-waste using redox active tetrathiotungstate ligands

Marie A. Perrin [1], Paul Dutheil [1,2,3], Michael Wörle[1] & Victor Mougel [1] ✉

Rare-earth elements (REEs) are critical to our modern economy, yet their mining from natural ores bears a profound environmental impact. Traditional separation techniques are chemical and energy-intensive because their chemical similarities make REEs very challenging to purify, requiring multiple extraction steps to achieve high purity products. This emphasizes the need for sustainable and straightforward separation methods. Here we introduce a strategy for the direct separation of europium (Eu) from complex mixtures under ambient conditions, leveraging on the redox non innocence of purely inorganic tungsten tetrathiolate ($WS_4^{2-}$) ligands. The recovery of Eu is achieved upon reduction of Eu(III) to a Eu(II) coordination polymer, driven by an induced internal electron transfer from the tetrathiotungstate ligand. Applying this strategy to unconventional feedstock such as spent energy-saving lamps allows selective europium recovery with separation factors over 1000 and recovery efficiency as high as 99% without pre-treatment of the waste.

The efficient utilization of natural resources is pivotal for constructing a sustainable and circular economy. However, the mining of mineral resources, vital to both the digital and the energy transition, currently incurs substantial societal and environmental consequences. Unlike the high recycling rates of aluminum and iron, the recovery rate of rare-earth elements (REEs) remains under 1%. Listed as critical minerals by the International Energy Agency, REEs are essential to numerous technologies, ranging from wind turbines to electric cars, optical displays, and energy-saving lamps. Meeting the current energy transition goals implies a sevenfold increase in REEs demand by 2040, a challenge exacerbated by the geopolitical dynamics surrounding their production[1,2]. These issues highlight the need for developing sustainable and straightforward methods to recover these elements from unconventional feedstocks[3,4]. Electronic waste is a significant yet overlooked reserve of REEs and could potentially replenish the market while being devoid of geographical constrains[5–7]. Among REEs, Europium (Eu) stands out due to its scarcity, with its concentration in common ores ranging from just 0.05−0.10% w/w, justifying its presence in the 2022 list of critical minerals of the US Geological Survey and 2023 list of Critical Raw Materials of the EU[8,9]. Historically, Europium's primary application was as phosphor in fluorescent lamps,

contributing to high market prices for Eu. However, as these lamps are gradually phased out, market prices have declined, rendering current recycling methods economically unviable at an industrial scale[10]. Yet, the development of more efficient separation strategies for Eu presents an appealing opportunity, given the abundance of inexpensive, spent fluorescent lamps waste, as their REEs content is significantly high from an extraction perspective−around 230 kg per ton, about 17 times higher than in natural ores[11,12]. Separation processes for REEs by conventional methods (*e.g.* solvent extraction, ion exchange) are both time and resource intensive. On the other hand, redox chemistry has been often chosen as a method of choice for europium separation and recovery, exploiting the fact that Eu possesses the highest reduction potential among REEs ($E_{1/2} = -0.34$ V *vs*. SHE for the Eu(II/III) couple)[13]. However, current processes rely on the use of strong sacrificial reducing agents or prolonged UV illumination[14–16].

In the present work, we aimed to explore the use of fully inorganic, redox non-innocent ligands that could be triggered by an external stimuli to induce Eu(III) reduction and subsequent separation from complex lanthanide mixtures (Fig. 1a). The redox-non innocence of ligands is a widely observed phenomenon in enzymatic systems, and has been shown to enable kinetically controlled separation of REEs[17].

[1]Department of Chemistry and Applied Biosciences, ETH Zürich, Vladimir-Prelog-Weg 1-5, 8093 Zürich, Switzerland. [2]Laboratory of Radiochemistry, Nuclear Energy and Safety Division, Paul Scherrer Institute, Forschungsstrasse 111, Villigen, PSI CH-5232, Switzerland. [3]Department of Radiation Safety and Security, Paul Scherrer Institute, Forschungsstrasse 111, Villigen, PSI CH-5232, Switzerland. ✉e-mail: mougelv@ethz.ch

**a** Current separation process by solvent extraction

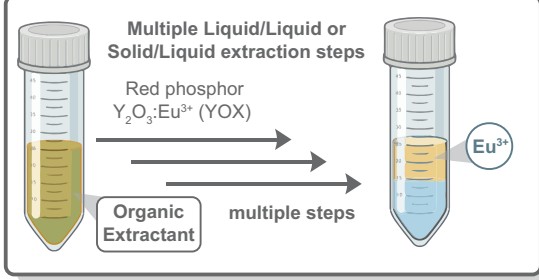

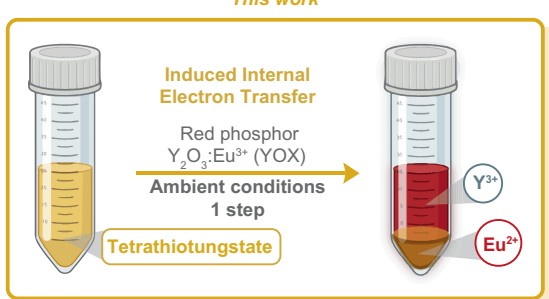

**b** Proposed recycling process for europium from compact fluorescent lamps

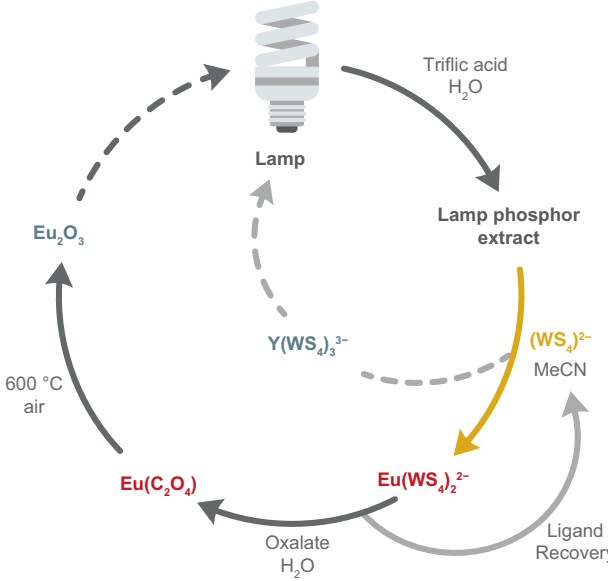

**Fig. 1 | Selective precipitation of europium and application to the recycling of lamp phosphor powder. a** Liquid–liquid extraction process for REEs separation versus one-step precipitation using $WS_4^{2-}$ ligands. Created with BioRender.com

released under a Creative Commons Attribution-NonCommercial-NoDerivs 4.0 International license. **b** Circular process from the recovery of europium from compact fluorescent lamps.

Inspired by the redox-non innocence of sulfur-containing ligands observed in a large number of metalloenzymes and metallocofactors[18–20], we targeted here the investigation of redox non innocent sulfur-containing ligands for rare earth complexes. Sulfur claims a special role in ligand redox non-innocence thanks to its unique redox properties: reduced sulfide ions ($S^{2-}$), are reasonably strong reducing agents, which can convert to disulfide ($S_2^{2-}$) or elemental sulfur while concurrently reducing the metal center they coordinate[21]. These redox properties can be enhanced by a synergistic interaction with the ligands incorporating these sulfur ions, or the metal centers they coordinate. A unique example of a strong redox interplay between sulfur and metal centers is provided by tetrathiometallates anions of early transition metals ($MS_4^{2-}$ with M = V(V), Mo(VI), W(VI), Re(VII)). These fully oxidized $d^0$ metal centers can nevertheless act as strong reducing agents in the presence of an external oxidizing agent[22]. Such counter-intuitive reactions, where an oxidation triggers a reduction, are named induced internal electron transfers (IIET) or redox-induced electron transfers[23,24]. In the present case, it has been proposed that the oxidation of one sulfide of the tetrathiometallate induces the formation of $S_2^{2-}$ by oxidation of a second sulfide ligand, concomitantly resulting in the reduction of the metal center[22,25–27]. In addition to their redox properties, tetrathiometallates display a very versatile coordination chemistry and can bind to other metal centers via bridging sulfide ions, stabilizing low-valent transition metals[28–30]. But despite the very rich coordination chemistry of these ligands[22,28–30], we identified only two reports involving REEs, both combining tetrathiometallates with lanthanocene precursors[31,32].

Here, we report the use of tetrathiotungstate anions as fully inorganic, redox non-innocent ligands for the preparation of trivalent europium and yttrium complexes. We exploit their induced internal electron transfer properties to trigger the reduction of Eu(III) to Eu(II), resulting in the formation of a highly insoluble Eu(II) tetrathiotunsgtate coordination polymer upon mild external stimuli, such as ambient light or mild heating. This strategy enables a facile and highly selective separation of europium from complex lanthanide mixtures such as end-of-life compact fluorescent lamps (Fig. 1b).

## Results

### Synthesis and characterization

Treating an excess of $(NEt_4)_2WS_4$ in MeCN (3 equiv.) with $Eu(OTf)_3$ (1 equiv.) resulted in an immediate color change, from bright yellow to dark red. Under ambient light and at room temperature, a golden-brown precipitate rapidly formed, accompanied by a shift in the UV–vis spectrum of the supernatant (Supplementary Fig. 6). The high insolubility of the resulting precipitate precluded its recrystallization for single-crystal X-ray diffraction (XRD) analysis. However, slow diffusion between the reactants allowed us to isolate single crystals of a coordination polymer with the formula $[NEt_4]_2[Eu^{II}(WS_4)_2]$ (**1**) which was characterized by XRD (Fig. 2a). The bond valence sum analysis of the complex corroborated the divalent state of the Eu center, a finding which was further substantiated by magnetic susceptibility measurements and X-ray Photoelectron Spectroscopy (XPS) (Supplementary Fig. 16). Respectively, the tungsten centers remained in the +6 oxidation state, as highlighted by XANES measurement of complex **1** at the W $L_{III}$-edge, showing identical white line energy with the free $WS_4^{2-}$ ligand (Supplementary Fig. 19). Despite displaying comparable elemental analyses, the powder X-ray diffraction pattern of the golden-brown precipitate did not match with that expected from the single crystal structure analysis of **1**. Suspecting different polymorphs of this coordination polymer, we measured X-ray total scattering of the precipitate, and obtained its pair distribution function (PDF) (Supplementary Fig. 24). The PDF revealed the expected short-range ordering and suggest that the coordination geometry for the W and Eu ions was retained and is indeed similar to the one determined from the single crystal data, further confirming the close relation of the precipitate and the single crystalline phase of **1**.

Interestingly, when the same reaction was conducted at −35 °C in the absence of light, the solution retained its dark red color, and no precipitate was observed, underscoring the critical role of synthesis conditions in stabilizing Eu in its trivalent state. Layering this dark red solution with $Et_2O$ yielded dark red crystals of the trivalent complex $[NEt_4]_3[Eu^{III}(MeCN)_2(WS_4)_3]$ (**2**) (Fig. 2a). In situ Raman spectroscopy monitoring of complex **1** synthesis revealed the emergence of a new

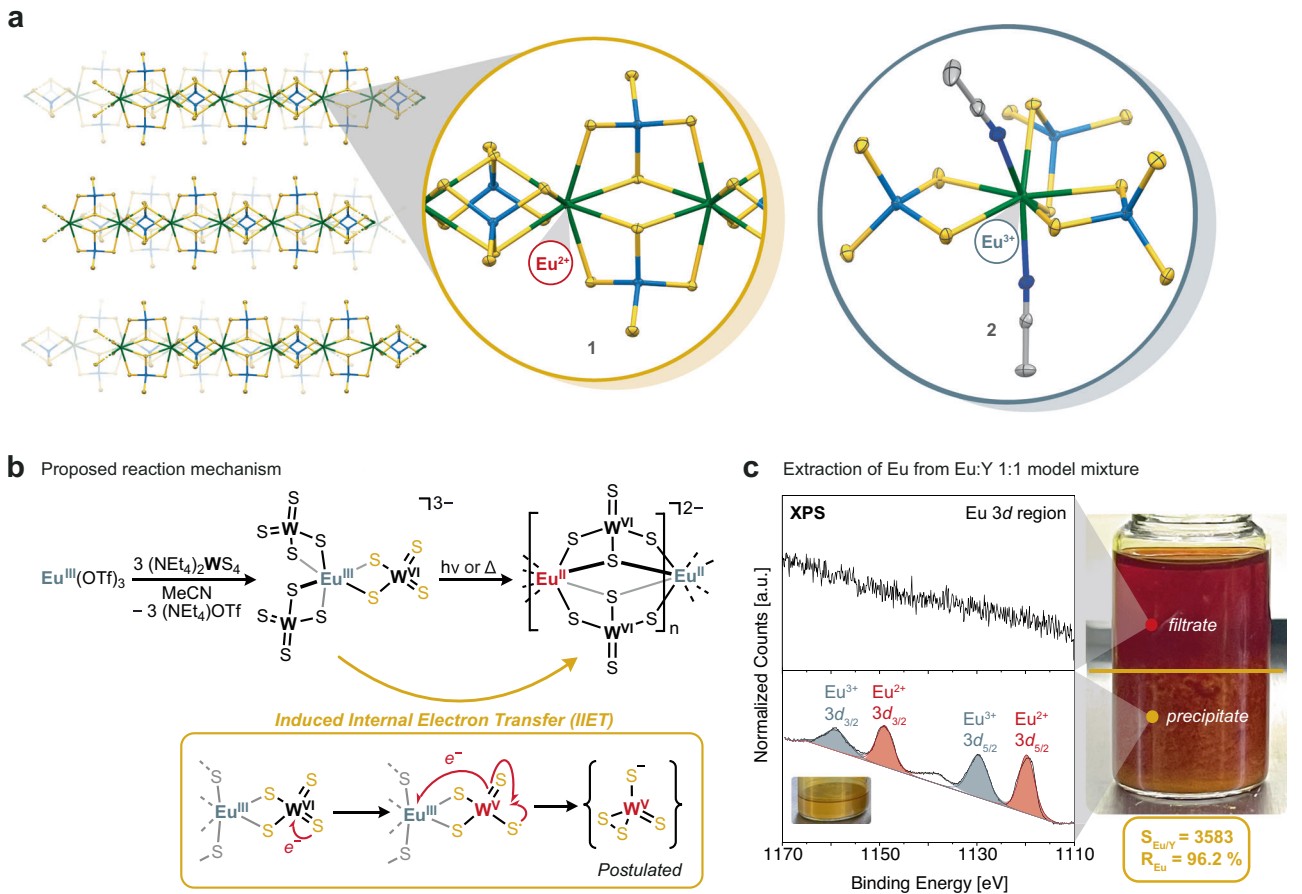

**Fig. 2 | Internal electron transfer from tetrathiotungstate ligand inducing the reduction and precipitation of europium under ambient conditions. a** Crystal structures of complex **1** (left) and **2** (right). Ellipsoids are drawn at the 50% probability level and all hydrogen atoms, tetraethylammonium countercations and lattice solvent molecules were omitted for clarity. Atom (color): Eu (green), W (light blue), S (yellow), N (dark blue), C (gray). **b** Proposed induced internal electron transfer mechanism for the formation of the Eu(II) coordination polymer. **c** Separation of a 1:1 (wt%) Eu:Y mixture and associated XPS spectra of the Eu 3$d$ region for the filtrate (top) and precipitate (bottom).

lower symmetry W-S vibration at 491 cm$^{-1}$ immediately after mixing the two reactants, indicative of the fast coordination of the tetra-thiotungstate ligand, while the appearance of a vibration above 500 cm$^{-1}$ after 10 minutes suggests the formation of a disulfide complex (Supplementary Fig. 8)[33], highlighting the ligand's involvement in a redox process. This observation, coupled with the mild reduction potential of Eu(III) to Eu(II) exhibited in the cyclic voltammogram of **2** ($E_{1/2}$ = − 0.55 V vs. Fc/Fc$^+$, Supplementary Fig. 1), suggests the role of the tetrathiotungstate ligand in promoting an internal electron transfer mechanism, initiated by an external stimulus such as light or heat, which likely triggers the reduction of europium and the formation of **1**, as shown in Fig. 2b.

**Selective europium precipitation from model mixtures**

The formation of a highly insoluble precipitate as a result of the reduction from Eu(III) to Eu(II) under mild external stimuli, together with the high solubility of the trivalent complex paved the way for a potential separation process. In extending our study to yttrium, commonly found alongside europium in phosphors for electronic displays and energy-saving lamps, we observed the formation of the highly soluble complex [NEt$_4$]$_3$[Y$^{III}$(MeCN)(WS$_4$)$_3$] (**3**) (Supplementary Fig. 23). The separation was therefore investigated using a 1:1 (wt %) model mixture of Eu(OTf)$_3$ and Y(OTf)$_3$ in the presence of an excess of (NEt$_4$)$_2$WS$_4$ (8 equiv.). Upon stirring at room temperature under ambient light, a large amount of golden-brown precipitate was

observed after an hour. The solid was isolated by centrifugation and the resulting red filtrate was taken to dryness to yield a bright red powder. Characterization of both solids by XPS suggests that the precipitate contains mainly europium while in the red filtrate mainly yttrium is found (Fig. 2c). The average separation factor was determined by Inductively Coupled Plasma Optical Emission Spectroscopy (ICP-OES), according to Eq. (1), to be $S_{Eu/Y}$ = 3583 ± 1803 for a 1:1 (wt%) Eu:Y mixture and $S_{Eu/Y}$ = 2888 ± 55 when starting from the 1:14 (Eu:Y) ratio typically found in commercial phosphors (Supplementary Table 8 and 10). To the best of our knowledge, these values outperform separation factors for all methods reported to date, which usually range between 2 to 100 for Eu:Y (Supplementary Table 9)[11].

**Europium recovery from spent energy-saving lamps**

Such high separation factors encouraged us to investigate the separation of Eu in more complex mixtures such as fluorescent lamps (Fig. 3 and Supplementary Movie 1). The phosphors were directly extracted from spent fluorescent lamps using aqueous tri-fluoromethanesulfonic acid, yielding, after filtration from the glass pieces and vacuum drying at 200 °C, a light gray powder (3.3 wt% from the crushed lamp material). This extraction step, which is critical to the efficiency of our process, uses trifluoromethanesulfonic acid because of its unique ability to dissolve phosphors in a single step, but more importantly, because the weakly coordinating triflate anions that interact with the lanthanide ions are essential for efficient

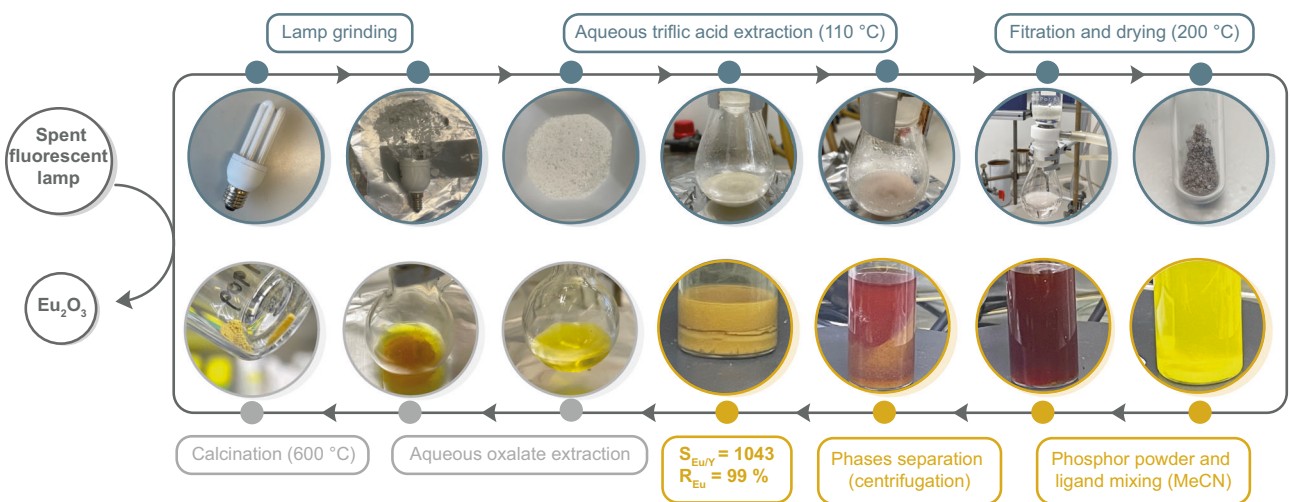

**Fig. 3 | Illustrated circular process for europium recovery from a spent compact fluorescent lamp according to the process described in the present work.** The 14 photos illustrate the main steps of the process, namely the separation of europium in a single step from the triflic acid extract of a lamp powder using tetrathiotungstate, the extraction of Eu(II) from the ligand using oxalate in water and its subsequent calcination to yield europium oxide. This process is further illustrated in the Supplementary Movie 1.

**Table 1 | Separation factors for Eu/Y separation from phosphor powders with different methods**

| Reference | Method | Extractant | Eu recovery efficiency (%) | Separation factor $S_{Eu/Y}$ |
|---|---|---|---|---|
| Rabah[38] | Liquid–liquid extraction | Benzyltrimethylammonium chloride | 99 | 9.5 |
| Binnemans et al. (2019)[39] | Ionic liquid | [C101][SCN] | 98 | 17.6 |
| Bertau et al. (2021)[40] | Liquid–liquid extraction | Cyanex 923 | | 20.49 |
| Patil et al. (2021)[12] | Liquid–liquid extraction | HDEHP | – | 35 |
| Van den Bogaert et al. (2015)[15] | Photochemical reduction | $(NH_4)_2SO_4$ | 50 | 48 |
| Tunsu et al. (2016)[41] | Liquid–liquid extraction | Cyanex 572 | – | 61 |
| Wu et al. (2019)[16] | Photochemical reduction | $(NH_4)_2SO_4$ | 61.2 | 61 |
| This work | IIET | $(NEt_4)_2WS4$ | 98.9* ± 0.8 | 1043[a] ± 322 |

[a]Average value from duplicate experiments—see Supplementary Table 13. The separation factors and the recovery efficiency were determined according to Eqs. (1) and (2), respectively.

ligand displacement upon addition of $WS_4^{2-}$ in the subsequent step. Analysis by ICP-OES revealed a composition of 0.93 wt% europium and 12.6 wt% yttrium, defining a Eu:Y ratio of 1:13.6, well in the range of typical $Y_2O_3$:$Eu^{3+}$ phosphor compositions[11]. Similarly to the behavior observed with the model mixture, the addition of this solid extract to a solution of $(NEt_4)_2WS_4$ in MeCN resulted in an immediate color change from bright yellow to dark red, followed by the formation of a golden-brown precipitate. Characterization of both the precipitate and the supernatant by ICP-OES underscored the very good separation of Eu from Y, resulting in a benchmark separation factor $S_{Eu/Y} = 1043 \pm 322$ and an efficiency of Eu removal of $98.9 \pm 0.8\%$ (Eq. (2), Supplementary Table 13). Such a high separation factor, over an order of magnitude higher than the best reported ones, is particularly impressive given the fact that the extract powder was obtained in a single step from the lamp, with no subsequent treatment (Table 1).

### Circular process for europium recovery and ligand recycling
To further investigate the relevance of using tetrathiotungstate as an extractant for europium recycling, we attempted to close the loop via the recovery of the tetrathiotungstate ligand and the isolation of a europium oxide phase. The treatment of an aqueous solution of the golden solid with ammonium oxalate afforded a red-brown precipitate (43.21 wt% of Eu). This precipitate was then calcined under air at 600 °C for 2 h, resulting in the formation of $Eu_2O_3$. Relatedly, UV–vis spectroscopy of the yellow supernatant, as obtained after filtration of

the precipitate, revealed the presence of a single thiotungstate species, $WS_4^{2-}$ (Supplementary Fig. 7). This circular treatment could also be applied to the 1:1 (wt%) Eu:Y separation from the model mixture, yielding $Eu_2O_3$ with 90% purity (Fig. 1b).

This study demonstrates the unparalleled capability of fully inorganic tetrathiotungstate ligands to facilitate a highly selective, one-step recovery of europium from end-of-life materials such as lamp phosphors. Owing to its straightforward methodology and cost-effectiveness, we anticipate that this approach will attract interest for a wider array of applications and sources of rare-earth elements.

## Methods
### General methods
Unless stated otherwise, syntheses were carried out under strict inert Argon atmosphere using Schlenk techniques or inside Vigor® gloveboxes. Solvents were dried using a Vigor® solvent purification system. Diethyl ether was additionally dried over potassium/benzophenone, distilled, degassed by 3 freeze-pump-thaw cycles and stored over 4 Å molecular sieves for at least 3 days prior to use. Likewise, acetonitrile was degassed by 3 freeze-pump-thaw cycles and stored over 3 Å molecular sieves prior to use. Elemental analyses were carried out at the Molecular and Biomolecular Analysis Service (MoBiAS) of ETH Zürich on a LECO TruSpec® Micro spectrometer. UV–Vis electronic absorption data in MeCN were collected on an Agilent Cary 60 UV–Vis Spectro- photometer, connected to a sampling probe ($d = 2$ mm) located inside a Glovebox via an optical fiber.

X-ray Crystallography data were collected on a Rigaku XtaLAB Synergy-S diffractometer equipped with a HyPix-6000HE detector using CuKα radiation ($\lambda = 1.54184$ Å) at 100 K. After data collection, structures were solved by intrinsic phasing (SHELXT) and refined by full-matrix least-squares procedures on $F^2$ using SHELXL in the olex2 program suite[34–36]. All non-hydrogen atoms were refined with anisotropic displacement parameters. The hydrogen atoms were placed in positions of optimized geometry. Magnetic susceptibility measurements in the solid state were carried out on a Gouy Balance (Johnson Matthey). X-ray photoelectron spectroscopy (XPS) measurements were performed on a Sigma II instrument (Thermo Electron) equipped with an Alpha 110 hemispherical analyzer. X-ray Powder diffraction data suitable for Pair Distribution Function (PDF) analysis were collected from a sample sealed in Mark-tube (0.5 mm diameter) using a Stoe STADI P diffractometer (AgKα1 radiation, $\lambda = 0.55941$ Å, curved Ge-monochromator) equipped with a Mython 4 K detector. Raman spectroscopy data were collected using a Thermo Scientific DXR Smart Raman spectrometer and processed with the OMNIC software. Cyclic voltammograms were recorded under strictly anaerobic conditions in a conventional three-electrode single-compartment cell (20 mL) using glassy carbon as working electrode (diameter 3 mm), a platinum wire as counter-electrode and a silver wire dipped in a 0.01 M solution of AgNO$_3$ in a 0.1 M solution of TBAPF$_6$ in MeCN as a reference electrode. The reference electrode was separated from the cell using a guard filled with the same electrolyte as used in the cell, separated by a Vycor® frit. The potential was controlled by a BioLogic SP-300 potentiostat (Bio-Logic Science Instruments SAS). Inductively Coupled Plasma Optical Emission spectroscopy (ICP-OES) measurements were carried out with an Agilent 5110 instrument (Agilent Technologies, Inc.) equipped with a double-pass spray chamber and a SeaSpray concentric glass nebulizer. During the measurements, the instrument was operated in its axial mode, with 0.7 L min$^{-1}$ nebulizer flow (Ar), 12 L min$^{-1}$ plasma flow (Ar), 1 L min$^{-1}$ auxiliary flow (Ar), and 1.2 kW RF power. The instrument was calibrated externally from single-element standards (1000 mg L$^{-1}$, TraceCERT ®, Merck KGaA, Germany) and their dilutions, allowing Eu and Y to be quantified with the 381.967 nm and 360.074 nm spectral lines, respectively. The samples were prepared by dissolution in analytical grade concentrated HNO$_3$ and a subsequent dilution with milliQ water (18.2 mΩ). The separation factor $S_{Eu/Y}$ was determined by the following equation (Eq. 1):

$$S_{Eu/Y} = D_{solid} \times D_{filtrate} = \frac{\eta_{Eu}}{\eta_Y} \times \frac{\eta_Y}{\eta_{Eu}} \tag{1}$$

Where the weight ratios (wt%) were determined by ICP-OES spectroscopy.

The recovery efficiency was determined as the percentage of Europium extracted from the original mixture using the following equation (Eq. 2):

$$R_{Eu} = 100 - \left[ \frac{C_{Eu-liquid}}{C_{Eu-imput}} \times 100 \right] \tag{2}$$

where the concentrations were determined by ICP-OES spectroscopy.

### Synthesis of [NEt$_4$]$_2$[Eu$^{II}$(WS$_4$)$_2$] (1)
In a 25 mL scintillation vial, (NEt$_4$)$_2$WS$_4$ (200 mg, 0.35 mmol, 3 equiv.) was solubilized in 5 mL of MeCN, resulting in a bright yellow solution. Eu(OTf)$_3$ (70 mg, 0.117 mmol, 1 equiv.) was then added, resulting in an immediate color change to dark red. After 1 h a large amount of a golden-brown precipitate could already be observed. The reaction was further stirred at room temperature for 24 h, before the solution was centrifuged and washed with MeCN (3 × 2 mL) (118 mg, 0.113 mmol, 96% yield). Single crystals suitable for XRD analysis were obtained from

the slow diffusion of the reactants at room temperature by layering dilute solutions under ambient conditions. Alternatively, single crystals of 1 could also be obtained by slow vapor diffusion of Et$_2$O onto a solution of the reactants. UV−Vis (MeCN, 2 × 10$^{-4}$ M, 2 mm path) $\lambda_{max}$ (nm) ($\epsilon$ (M$^{-1}$ cm$^{-1}$)): 287 (13 559); 398 (17 952); Elemental analysis found (calc.)% for 1 (C$_{16}$H$_{40}$EuN$_2$S$_8$W$_2$): C, 18.23 (18.54); H, 3.78 (3.89); N, 2.76 (2.70). A magnetic susceptibility of 7.2 μ$_B$ was determined at room temperature, in good agreement with the total spin of 7/2 (μ = 7.9 μ$_B$) expected for Eu(II)[37].

### Influence of light and heat here on the formation of (1).
To investigate the impact of external stimuli (heat and light) on the formation of (1), the same reaction than described above was carried out in the absence of light at room temperature and at 60 °C. For each reaction (no light and ambient temperature, ambient light and temperature, ambient light and 60 °C) Eu(OTf)$_3$ (11 mg, 0.018 mmol, 1 equiv.) and (NEt$_4$)$_2$WS$_4$ (29 mg, 0.050 mmol, 2.7 equiv.) were solubilized in 2 mL of MeCN. To better discriminate the influence of these stimuli, the precipitate formed over the reaction was collected, dried and weighted after 1 h stirring. The synthesis under ambient light at room temperature yielded 9.46 mg of 1 were collected (51% yield), a decreased yield was observed when the reaction is conducted in the dark (6.4 mg of 1 collected, 35% yield), and an increased yield was determined when the reaction is conducted at 60 °C (15.9 mg of 1 collected, 83% yield).

### Synthesis of [NEt$_4$]$_3$[Eu$^{III}$(MeCN)$_2$(WS$_4$)$_3$]·MeCN (2)
The synthesis was conducted with a minimal exposure to ambient light and always handling solutions cooled at −35 °C. In a 25 mL scintillation vial, (NEt$_4$)$_2$WS$_4$ (200 mg, 0.350 mmol, 3 equiv.) was solubilized in 5 mL of MeCN resulting in a bright yellow solution. The latter was treated with Eu(OTf)$_3$ (70 mg, 0.117 mmol, 1 equiv.) resulting in an immediate color change to dark red. The reaction was kept at −35 °C for 24 h before it was layered in the dark with pentane (2 mL) and Et$_2$O (5 mL) and further stored at −35 °C. After a day, dark red crystals were isolated by pipetting out the supernatant. The large red crystals co-crystalize with small amounts of the fine golden-brown precipitate of 1, but this fine powder can be separated from the larger single crystals of 2 by suspending it in Et$_2$O (3 × 3 mL). Large red crystals of the title compound obtained after that rinsing step were dried under vacuum (122 mg, 0.077 mmol, 66% yield). UV−Vis (1.2 × 10$^{-4}$ M MeCN, 2 mm path) $\lambda_{max}$ (nm) ($\epsilon$ (M$^{-1}$ cm$^{-1}$)): 285 (42 147); 398 (41 713); 440 (64 232). $E_{1/2} = -0.55$ V (vs. Fc/Fc$^+$) in 0.1 M TBAPF$_6$ in MeCN. Elemental analysis found (calc.)% for 2·0.5 MeCN (C$_{29}$H$_{67.5}$EuN$_{5.5}$S$_{12}$W$_3$): C, 22.01 (22.02); H, 4.74 (4.30); N, 4.53 (4.87).

### Synthesis of [NEt$_4$]$_3$[Y$^{III}$(MeCN)(WS$_4$)$_3$] (3)
In a 25 mL scintillation vial, (NEt$_4$)$_2$WS$_4$ (100 mg, 0.174 mmol, 3 equiv.) was solubilized in 5 mL of MeCN, resulting in a bright yellow solution. Y(OTf)$_3$ (31.2 mg, 0.058 mmol, 1 equiv.) was added as a solid to this solution, resulting in an immediate color change to bright orange. The reaction was stirred at room temperature for 24 h before it was layered with pentane (2 mL) and Et$_2$O (5 mL) and crystallization was set at −35 °C. After a day, red crystals were isolated by pipetting out the supernatant, washed with Et$_2$O (2 × 2 mL) and dried under vacuum (82 mg, 0.055 mmol, 95% yield). UV−Vis (1 × 10$^{-4}$ M MeCN) $\lambda_{max}$ (nm) ($\epsilon$ (M$^{-1}$ cm$^{-1}$)): 220 (39 335); 283 (51 020); 398 (44 689); 441 (6 079). Elemental analysis found (calc.)% for 3·0.5 MeCN (C$_{27}$H$_{64.5}$N$_{4.5}$S$_{12}$W$_3$Y): C, 22.23 (21.95); H, 4.81 (4.40); N, 3.90 (4.27).

### Separation of Eu from 1:1 Eu/Y model mixture
In a 20 mL scintillation vial, (NEt$_4$)$_2$WS$_4$ (190 mg, 0.336 mmol, 7 equiv.) was solubilized in 10 mL of MeCN, resulting in a bright yellow solution. Eu(OTf)$_3$ (30 mg, 0.050 mmol, 1 equiv.) and Y(OTf)$_3$ (30 mg, 0.056 mmol, 1.1 equiv.) were added as solids to this solution,

resulting in an immediate color change to dark red. After 1 h a golden-brown precipitate was observed. The reaction was stirred at room temperature for 24 h, before the solution was centrifuged to separate the red filtrate from the golden-brown precipitate. The latter was washed with MeCN (2 × 4 mL) and then both phases were dried under vacuum yielding a golden solid (46.6 mg) and a red solid (178.9 mg) which were characterized by ICP-OES ($S_{Eu/Y}$ = 3583, $R_{Eu}$ = 96%, average value from duplicates) (Supplementary Table 8). It should be noted that when the red filtrate is stored at −35 °C, single crystals of complex **3** can be isolated. Elemental analysis found for the golden solid (calc.)% for $C_{16}H_{40}EuN_2S_8W_2$ (Europium phase): C, 18.52 (18.54); H, 3.90 (3.89); N, 2.80 (2.70), in agreement with its assignment as complex 1.

### Separation of Eu from 1:14 (wt%) Eu/Y model mixture

In a 25 mL scintillation vial, $(NEt_4)_2WS_4$ (480 mg, 0.840 mmol, 50 equiv.) was solubilized in 20 mL of MeCN resulting in a bright yellow solution. $Eu(OTf)_3$ (10 mg, 0.016 mmol, 1 equiv.) and $Y(OTf)_3$ (140 mg, 0.261 mmol, 15.6 equiv.) were added as solids to this solution, resulting in an immediate color change to dark red. After one hour stirring at room temperature, a golden-brown precipitate had formed. The reaction was stirred at room temperature for 24 h, before the solution was centrifuged to separate the red filtrate from the golden-brown precipitate. The latter was washed with 2 ×4 mL of MeCN and then both phases were dried under vacuum yielding a golden solid (18.5 mg) and red solid (599.8 mg) which were characterized by ICP-OES ($S_{Eu/Y}$ = 2888, $R_{Eu}$ = 99%, average value from duplicates) (Supplementary Table 10). The composition of the golden precipitate was also monitored by ICP-OES, illustrating that best separation is reached after 24 h (Supplementary Table 11).

### Extraction of lamp phosphor

This experiment was carried out in a well-ventilated fumehood. A compact fluorescent light bulb (PHILIPS Genie 14 W energy saver 230-240 V) was crushed inside a plastic bag to separate the glass from the bulb socket. The extracted glass (21.150 g) was mortared into a fine powder. That powder was then suspended in a 1:2 mixture of trifluoromethanesulfonic acid (2 mL) in water at 0 °C, resulting in an off-white slurry. The suspension was heated at 110 °C for 2 h resulting in a color change to light pink. Water was added (10 mL) and the slurry was stirred for 30 min before being filtered and the filtrate taken to dryness at 200 °C for 25 h (1.2275 g, 5.8 wt%) and was characterized by ICP-OES (Supplementary Table 12). The process is visually illustrated in Supplementary Movie 1.

### Separation of Eu from lamp phosphor

In a 25 mL scintillation vial, $(NEt_4)_2WS_4$ (0.500 g, 0.87 mmol) was solubilized in 20 mL of MeCN resulting in a bright yellow solution. Lamp phosphor powder extracted according to the process described above (0.200 g) was added as a solid to this solution, resulting in an immediate color change to dark red. After one hour a golden-brown precipitate had formed. The reaction was stirred at room temperature for 24 h, before the solution was centrifuged to separate the resulting golden-brown powder which was further washed with MeCN (2 × 4 mL) and dried under vacuum (24.6 mg, 12 wt%) and a red filtrate which was taken to dryness (659.4 mg, >3 wt% loss). Both phases were characterized by ICP-OES ($S_{Eu/Y}$ = 1043, $R_{Eu}$ = 99%, average value from duplicates) (Supplementary Table 13). The process is visually illustrated in Supplementary Movie 1.

### Recovery of Europium oxide from complex 1:1 (wt%) Eu:Y model mixture

The golden solid obtained from the 1:1 (wt%) Eu:Y separation from the model mixture (115 mg, 0.11 mmol, 1 equiv.) was solubilized in 5 mL of degassed miliQ water resulting in a yellow solution. Ammonium oxalate (40 mg, 0.28 mmol, 2.5 equiv.) was added as a solid to the solution, leading immediately to the precipitation of a dark red-brown solid. The reaction was stirred at room temperature for 1 h before the solution was centrifuged. The yellow supernatant was transferred to a Schlenk flask and taken to dryness. The red-brown solid was rinsed with water (2 × 10 mL) and dried under vacuum (25 mg, 0.102 mmol, 93% yield assuming a composition as $EuC_2O_4$), before being calcined in a quartz tube furnace at 600 °C for 2 h under a flow of dry air (150 mL min$^{-1}$). This resulted in the formation of a white solid (10.3 mg, 0.03 mmol) Elemental analysis found (calc.)% for $Eu_2O_3$: Eu, 77.56 (86.36). Y 2.198 (0), corresponding to a 90% purity.

Further information on the synthesis and characterization of the complexes and on the extraction procedures is available in the Supplementary Information.

## Data availability

Crystallographic data for the structures reported in the Article have been deposited at the Cambridge Crystallographic Data Center, under deposition numbers CCDC 2312650 (1), 2312646 (**2**) and 2312677 (**3**). Copies of the data can be obtained free of charge via https://www.ccdc.cam.ac.uk/structures/. All other data supporting the findings of the study, including experimental procedures and characterization (CV, UV, Raman, IR, XPS, XAS, XRD, PDF, ICP) are available within the paper and its Supplementary Information. Source data are provided for this paper on the ETH Research Collection data base under the https://doi.org/10.3929/ethz-b-000670351.

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

## Acknowledgements

The authors would like to thank Prof. Dr. Patrick Steinegger and Dr. Martin Heule for fruitful discussions and for providing access to the ICP-OES and Dr. Thomas Nauser for insightful discussions. They would also like to acknowledge the Paul Scherrer Institut (PSI) for the provision of beamtime at the SuperXAS beamline (X10DA) of the Swiss Light Source (SLS), and Dr. Daniel F. Abbott, Amrita Singh-Morgan and Lok Nga Poon for their help during beamtime. V.M. and M.A.P. thanks ETH Zurich for funding via ETH research grant (ETH-44-19-1).

## Author contributions

M.A.P. conducted the experiments with support from P.D and M.W.. M.A.P., P.D, M.W., and V. M. analyzed data. M.A.P. and V.M. wrote the article with support from all authors. M.A.P. and V.M. designed the research and V.M. supervised the project.

## Funding

Open Open access funding provided by Swiss Federal Institute of Technology Zurich.

## Competing interests

M.P. and V.M. are sole inventors on a patent application filed by ETH Zürich in June 2023 based on this work (European patent application number EP23306022). The remaining authors declare no competing interests.
