## [Peer Review File · Nature Communications]

Recovery of Europium from E-Waste Using Redox Active Tetrathiotungstate LigandsREVIEWER COMMENTS

Reviewer #1 (Remarks to the Author):

Authors studied a new method to separate Eu from the complex mixture of the lamp phosphor extract. The main idea is the use of the redox active tungsten tetrathiolate. And results showed that the method is capable Eu/Y separation. There are some points that need to be addressed:

1. The proposed closed-loop recycle and reuse of phosphor in Fig. 3 may be oversimplified. For example, main challenges facing Y and Eu recovery from phosphor of spent fluorescent lamps are the acid extraction part and subsequent separation and purification processes. For example, instead of triflic acid, mostly inorganic acids, such as sulfuric acid or hydrochloric acid are used instead triflic acid as illustrated in the work. In that case, other metals including zinc, lead, and iron will be present. How will they interfere with the redox reactions? Besides, the acid extraction unit is energy-intensive, time consuming, and with a large carbon footprint. Relatively speaking, the separation of Y and Eu may not be the top issue to be tackled.
2. The method is effective in separating Y from Eu. Could authors elaborate on the advantages of the method in comparison with conventional ones, such as liquid-liquid extraction, membrane, and ion exchange?

Reviewer #2 (Remarks to the Author):

Rare earth recycling technology can recover rare earth elements from multiple sources, including electronic waste, mining waste, and industrial and consumer products. By recycling rather than directly extracting from minerals, it is possible to reduce environmental impact and decrease the demand for new resources. In this manuscript, the authors introduce a strategy for the direct separation of europium (Eu) from complex mixtures under ambient conditions, leveraging on the redox non-innocence of purely inorganic tungsten tetrathiolate (WS₄²⁻) ligands. The recovery of Eu is achieved upon reduction of Eu(III) to a Eu(II) coordination polymer, driven by an induced internal electron transfer from the tetrathiotungstate ligand. I would suggest accepting it after the following concerns are addressed.

1. Figure 2 is not clear, with some critical information being blurry, such as the valence state of Eu. It is recommended that the author redraws this Figure and ensures that all the relevant information is fully conveyed.
2. Given the volatility and toxicity of acetonitrile as a solvent, it is not a good choice. Can it be replaced with other solvents, such as water?
3. In this separation system, does the anion have any effect on the redox behavior of Eu? Would replacing the organic anion trifluoromethanesulfonate with common inorganic anions such as sulfate, nitrate, or chloride affect the separation efficiency?
4. Besides Eu³⁺, can tetrathiotungstate ligands also reduce other trivalent rare earth cations? The authors need to supplement with relevant experiments.
5. The valence state of W in the crystal structures need to be confirmed by relevant characterization

techniques.

6. In the synthesis of $[\text{NEt}_4]_2[\text{Eu}(\text{WS}_4)_2]$ (1), I found that 1 was obtained by slow diffusion, and by analyzing the structure of the precipitate that appeared during the synthesis process, it was consistent with 1, and I would like to ask if it is possible to obtain the crystal of 1 by reducing the concentration of the reaction and delaying the reaction time in the process of synthesizing 1.

7. I found that the conditions for the synthesis of 1 and 2 are basically the same, but 2 needs to be synthesized at $-35\text{ }^\circ\text{C}$ and protected from light, and the Eu in 2 is trivalent, what is the mechanism?

8. In the cyclic process of europium recovery and ligand recycling, ammonium oxalate is added to obtain a red-brown precipitate, can its structure be confirmed?

Response to Reviewers

A point-by-point answer to the reviewers' comments can be found below. For clarity, the latter appear in *bold/italics*, and our response is in normal text; modifications in the manuscript are highlighted in yellow.

Reviewer 1 comments:

1. "The proposed closed-loop recycle and reuse of phosphor in Fig. 3 may be oversimplified. For example, main challenges facing Y and Eu recovery from phosphor of spent fluorescent lamps are the acid extraction part and subsequent separation and purification processes. For example, instead of triflic acid, mostly inorganic acids, such as sulfuric acid or hydrochloric acid are used instead triflic acid as illustrated in the work. In that case, other metals including zinc, lead, and iron will be present. How will they interfere with the redox reactions? Besides, the acid extraction unit is energy-intensive, time consuming, and with a large carbon footprint. Relatively speaking, the separation of Y and Eu may not be the top issue to be tackled."

We are somehow surprised by the comment that the proposed loop is oversimplified but apologize for the confusion that this figure may have provided. We would like to clarify that this is not a hypothetical recycling loop, but the extraction process we carried out in the lab along the strategy defined in this paper, illustrated by 14 pictures taken along the way, as snapshot of the whole process (which is reported in the supporting video). As such, the scheme was not intended to describe existing procedures for the extraction of Ln ions from spent fluorescent lamps, but the specific strategy introduced here. In this work, we do not intend to integrate our strategy in existing extraction processes, using different acids than the one used in our study, but to develop an integrated extraction strategy, starting from the waste material, to the pure europium oxide. In the present case, our strategy uses triflic acid, and the selection of this acid allows the extraction of Europium with separation factors over 1000 in a single extraction step from the lamp waste, without any interference from the other metal present in the system and mentioned by the reviewer. This is significantly more energy efficient than classical Eu extraction strategies, showing much lower extraction factors. In particular, classical leaching strategies used prior to that work generally involve multiple leaching steps with inorganic acids of different concentrations, therefore generating consequential amounts of acidic waste. What we report is the use of a single leaching step which allows the rare earth extraction to be almost quantitative, therefore limiting the amount of waste generated.

We apologize if this was not clear enough in our initial version of the text, and we have now reformulated figure 3 caption to better illustrate that fact. It now reads:

"Fig. 3. Illustrated circular process for europium recovery from a spent compact fluorescent lamp according to the process described in the present work. The 14 photos illustrate the main steps of the process, namely the separation of europium in a single step from the triflic acid extract of a lamp powder using tetrathiotungstate, the extraction of Eu(II) from the ligand using oxalate in water and its subsequent calcination to yield europium oxide. This process is further illustrated in the supplementary video."

2. "The method is effective in separating Y from Eu. Could authors elaborate on the advantages of the method in comparison with conventional ones, such as liquid-liquid extraction, membrane, and ion exchange?"

We thank the reviewer for their comment. Indeed, comparing the effectiveness of our separation strategy with respect to those described in the literature is critical. As highlighted in the original version of the text, the method reported in this work possesses, to the best of our knowledge the highest separation factor $S_{Eu/Y}$ reported in the literature. As exemplified in table S14, the separation factor of our strategy using directly lamp phosphor powder sources is over 17 times higher than conventional liquid-liquid

extraction, ion-exchange and photochemical reduction methods. Due to their typical lower extraction factors, we had not initially reported other extraction strategies. We have now added the highest reported factors for each methods in Table S14 to better address the comment of the reviewer. We hope this now better illustrate the unique performance of our system.

Table S14. Separation factors from Y/Eu phosphor powders with different extractants.

Reference	Method	Extractant	Extraction steps	Eu recovery efficiency (%)	Separation factor $S_{Eu/Y}$
Rabah (2008) ²³	Liquid-liquid extraction	trimethyl-benzyl ammonium chloride	-	99	9.5
Binnemans et al. (2019) ²⁴	Ionic liquid	[C101][SCN]	4	98	17.6
Bertau et al. (2021) ²⁵	Liquid-liquid extraction	Cyanex 923	2		20.49
Patil et al. (2021) ²⁶	Liquid-liquid extraction	HDEHP	25	-	35
Van den Bogaert et al. (2015) ²⁷	Photochemical reduction	(NH ₄)SO ₄	1	50	48
Tunsu et al. (2016) ²⁸	Liquid-liquid extraction	Cyanex 572	10	-	61
Wu et al. (2019) ²⁹	Photochemical reduction	(NH ₄)SO ₄	1	61.2	61
This work	IIET	(NEt₄)₂WS₄	1	98.9*	1043*

*average value from duplicate experiments – see table S13

Reviewer 2 comments:

1. “Figure 2 is not clear, with some critical information being blurry, such as the valence state of Eu. It is recommended that the author redraws this Figure and ensures that all the relevant information is fully conveyed.”

We apologize for this lack of clarity. We have now simplified the figure and increased its quality to make sure no information is lost.

2. “Given the volatility and toxicity of acetonitrile as a solvent, it is not a good choice. Can it be replaced with other solvents, such as water?”

We are bit surprised by this comment from the reviewer regarding the toxicity of acetonitrile. According to its safety datasheet, acetonitrile possesses a LD₅₀ by ingestion of 2.64 g/kg, of over 2 g/kg by injection and over 16000ppm by inhalation. Its ingested toxicity is in the same order of magnitude that table salt (LD₅₀ = 3 g/kg), and its inhaled toxicity is in the same order of magnitude than ethanol (20000 ppm) (All LD₅₀ values were extracted from MSDS available online on the website of Fischer scientific, www.fishersci.com/us/en/catalog/search/sdshome.html). We hence believe that acetonitrile can be considered a very innocuous solvent and is in particular much less toxic than most solvents typically used in common extraction processes. With a boiling point of 82°C, it is also not a very volatile solvent and as all extraction steps are carried out in closed vessels the risk of inhaling is very limited. In addition, having the capability of evaporating the solvent under mild conditions (such as with acetonitrile) is important to maintain the energy efficiency of the whole process, as solvent removal is required to recover the Y phase.

Last, as illustrated in our recycling process, water induces the decoordination of the Eu ions, that we exploit to separate it from the tetrathiotungstate ligand. As such, it cannot be used instead of acetonitrile.

3. “In this separation system, does the anion has any effect on the redox behavior of Eu? Would replacing the organic anion trifluoromethanesulfonate with common inorganic anions such as sulfate, nitrate, or chloride affect the separation efficiency?”

We have shown in preliminary experiments that common inorganic acids such as proposed here do not enable an efficient separation and only generate ill-defined soluble mixtures, that we did not manage to crystalize, and no precipitate was formed. On the basis of these observations, we postulated that the use of a weakly coordinating anion is key to the extraction efficiency and choose using triflic acid accordingly. We have now added a short statement to the main text to highlight the importance of the triflic acid extraction step, and hope this clarifies this point better:

“The phosphors were directly extracted from spent fluorescent lamps using aqueous trifluoromethanesulfonic acid, yielding, after filtration from the glass pieces and vacuum drying at 200°C, a light grey powder (3.3 wt% from the crushed lamp material). This extraction step, which is critical to the efficiency of our process, uses trifluoromethanesulfonic acid because of its unique ability to dissolve phosphors in a single step, but more importantly, because the weakly coordinating triflate anions that interact with the lanthanide ions are essential for efficient ligand displacement upon addition of WS₄²⁻ in the subsequent step.”

4. “Besides Eu³⁺, can tetrathiotungstate ligands also reduce other trivalent rare earth cations? The authors need to supplement with relevant experiments.”

Under ambient conditions, only Eu can be reduced by the tetrathiotungstate ligands. We prepared and isolated complexes of the entire Ln series, and all of them remain trivalent. We have enclosed that series as an information for reviewer only, as we believe disclosing the whole series is out of the scope of the present paper, as it does not have implications for the separation of Europium from e-waste, the main

topic of the present work. The whole series will be disclosed in a separate publication, focusing of the physical properties of these complexes.

5. “The valence state of W in the crystal structures need to be confirmed by relevant characterization techniques.”

We thank the reviewer for this suggestion and apologize we had not sufficiently explicated that point in our original manuscript. We have now provided the XAS (W L₃-edge), XPS and bond valence sum analysis of the complexes, all concurring to an oxidation state of +6 for the W centers in both Eu(II) and Eu(III) complexes. We have now added these in the ESI in sections 7, 8 and 11.

6. “In the synthesis of [NEt₄]₂[EuII(WS₄)₂] (1), I found that 1 was obtained by slow diffusion, and by analyzing the structure of the precipitate that appeared during the synthesis process, it was consistent with 1, and I would like to ask if it is possible to obtain the crystal of 1 by reducing the concentration of the reaction and delaying the reaction time in the process of synthesizing 1.”

We thank the reviewer for this suggestion. It is indeed possible to obtain single crystals of **1** from three different strategies: using a dilute solution of the reactants, by slow diffusion, or by letting a dilute solution of **2** stand at room temperature for 24h. We have now added these 3 routes to single crystals of **1** in the ESI.

7. “I found that the conditions for the synthesis of 1 and 2 are basically the same, but 2 needs to be synthesized at -35 °C and protected from light, and the Eu in 2 is trivalent, what is the mechanism?”

We thank the reviewer for their question regarding the synthesis conditions of complexes **1** and **2**. The necessity to synthesize **2** at -35 °C and shield it from light is crucial for maintaining the Eu in its trivalent state. This condition is essential to our mechanism, where the tetrathiotungstate ligand engages in thermal or photochemical processes to reduce Eu(III) to Eu(II), leading to the precipitation of complex **1**. This point is in fact central to our study, designed to showcase the ligand's role in the redox behavior of europium. While we discussed this point extensively in the original version of the text, we acknowledge from the reviewer comment for the need for greater clarity in our manuscript on this point and appreciate the opportunity to further highlight the importance of synthesis conditions in directing the outcome of the reaction. We have now rephrased the following paragraph in the main text :

“Interestingly, when the same reaction was conducted at -35 °C in the absence of light, the solution retained its dark red color, and no precipitate was observed, **underscoring the critical role of synthesis conditions in stabilizing Eu in its trivalent state.** Layering this dark red solution with Et₂O yielded dark red crystals of the trivalent complex [NEt₄]₃[Eu^{III}(MeCN)₂(WS₄)₃] (**2**) (Fig. 2a). *In-situ* Raman spectroscopy monitoring of complex **1** synthesis revealed the emergence of a new lower symmetry W-S vibration at 491 cm⁻¹ immediately after mixing the two reactants, indicative of the fast coordination of the tetrathiotungstate ligand, while the appearance of a vibration above 500 cm⁻¹ after 10 minutes suggests the formation of a disulfide complex (Fig. S6),³³ **highlighting the ligand's involvement in a redox process.** This observation, coupled with the mild reduction potential of Eu(III) to Eu(II) exhibited in the cyclic voltammogram of **2**, (E_{1/2} = -0.55 V vs. Fc/Fc⁺, Fig. S16), **suggests the role of the tetrathiotungstate ligand in promoting an internal electron transfer (IET) mechanism, initiated by an external stimuli such as light or heat, which likely triggers the reduction of europium and the formation of 1, as depicted in Fig. 2b.**”

8. “In the cyclic process of europium recovery and ligand recycling, ammonium oxalate is added to obtain a red-brown precipitate, can its structure be confirmed?”

The identity of the precipitate observed after addition of ammonium oxalate could be confirmed by both FTIR and XPS spectroscopy, and perfectly matches literature data. We have now added these data to

the ESI of the paper (Figure S9 and S18). As for its structure, unfortunately, to the best of our knowledge, no single crystal XRD structure has been reported for divalent europium oxalate, and the obtained precipitate is amorphous (no signal observed in powder XRD spectra, in analogy with literature sample).

REVIEWERS' COMMENTS

Reviewer #1 (Remarks to the Author):

The significance of the work is the development of a new, integrated process for REEs recovery from waste lamps. It shows advantages as compared with traditional methods consisting of acid leaching using inorganic acids, followed by liquid-liquid extraction. It will potentially contribute to the field and related fields.

Reviewer #3 (Remarks to the Author):

The author has carefully revised the manuscript according to the comments, and I am satisfied with this revision.